# OpenReview forum: "Holistic Agent Leaderboard: The Missing Infrastructure for AI Agent Evaluation"
_ICLR.cc/2026/Conference — ICLR 2026 Poster_

### Official Review · Reviewer_YJJ9 · 2025-10-27

**Soundness:** 2
**Presentation:** 2
**Contribution:** 3
**Rating:** 2
**Confidence:** 4

**Summary:**

This paper introduces the Holistic Agent Leaderboard (HAL), a new infrastructure for evaluating AI agents. The authors identify key challenges in existing agent evaluation, such as lack of standardization, high costs, slow speed, and the failure to detect "cheating" or unsafe behaviors. HAL's contributions are threefold: (1) A standardized, open-source evaluation framework that uses distributed scheduling to accelerate evaluation from weeks to hours and automatically tracks costs. (2) A large-scale, three-dimensional (model, benchmark, framework) empirical study ($40k cost, 21,730 runs) that yields insights, such as the existence of a steep cost-accuracy Pareto frontier and the finding that increased reasoning effort can hurt performance. (3) An automated log analysis component using an LLM-assisted tool (Docent) to uncover failure modes, shortcuts, and unsafe behaviors missed by traditional accuracy metrics.

**Strengths:**

Significance and Scale: The paper addresses a good problem: the lack of robust, standardized, and holistic evaluation for AI agents. The scale of the evaluation (21,730 runs, $40k cost, 9 models, 9 benchmarks) is impressive and provides a valuable snapshot of the current agent landscape.

Holistic Approach: The move beyond simple accuracy to systematically include cost (via Pareto frontiers) and reliability (via log analysis) is a major strength. This is a crucial direction for making agent evaluation more meaningful for real-world deployment.

Compelling Log Analysis Insights: The results of the log analysis are good. Demonstrating that agents "cheat" (e.g., searching for answers on HuggingFace) or engage in catastrophic behaviors (e.g., using the wrong credit card) provides powerful, concrete evidence for the paper's main thesis.

**Weaknesses:**

Contribution is Primarily Engineering: The core contribution of this paper is heavily weighted towards engineering. While this is a valuable service, the paper struggles to isolate a clear, novel research contribution. The insights derived (e.g., "scaffolding matters", "cost-performance tradeoffs exist") are useful but not deeply surprising. The main product is the framework itself.

Lack of Technical Depth in Main Paper: The paper relegates too many implementation details. Key components like the distributed scheduling system, the cost-tracking mechanism, and the "lightweight modification" required to integrate new agents are not sufficiently detailed. This makes it difficult to assess the novelty of the engineering work and the reproducibility of the system from the paper alone.

Log Analysis Lacks Methodological Novelty: The automated log analysis, while a key part of the "holistic" claim, is not a new method. It is an application of an existing tool, Docent (Meng et al., 2025). The novelty lies in the scoring criteria the authors developed for Docent, but these are not discussed in depth, making it hard to assess their generality, development cost, or potential biases.

Missing Artifact Details: For a paper centered on an infrastructure contribution, the provided text does not contain clear information on where to access the open-source framework or the 2.5 billion token log dataset. The value of this paper is almost entirely tied to the community's ability to use and build upon this artifact, yet access to it is not made clear. This omission makes it difficult to evaluate the paper's primary contribution.

**Questions:**

Please solve the listed weakness above.

---

> ### Author Response · Authors · 2025-11-24
> **Additional analysis, responses to reviewer concerns**
>
> Thank you for taking the time to review our submission and we appreciate your feedback. We are glad that you found the significance, scale, and insights from HAL valuable.
>
> ## 1. Engineering-oriented contribution.
>
> Thanks for raising this concern. We appreciate the perspective.  That said, we do think there are several areas where the paper contributes novel methods, insights, and artifacts. We list some of these below, and have updated the paper PDF to clarify these contributions:
>
> - **Novel insights**:
>   - Many of the results we found were first reported in HAL, and couldn't have been reported without the large-scale system we built.
>   - For example, we comprehensively evaluate the effect of low vs. high reasoning across benchmarks, and find that higher reasoning does not help performance in the majority of cases. Given the strong claims about the impact of reasoning effort made by companies (such as GPT-5's results on Charxiv and SWE-Bench, and o3 and o4-mini's results on GPQA and AIME), we think it is important to test these claims empirically. Otherwise there is too much dependence on ''folk wisdom'' that isn't actually true and leads the community astray.
>   - We also found evidence of reliability failures through log analysis, such as agents refunding excessive amounts, charging the wrong credit card etc.
>   - We observe that increases in accuracy are not accompanied by decreases in token usage, on average. It is significant that stronger models are not displaying more efficient planning and execution.
>   - Our log analysis reveals new insights, including.
>     - the dramatic reliability failures in TauBench which would be catastrophic in deployment,
>     - detecting a bug in the official implementation of a popular benchmark's scaffold,
>     - stronger models succeed at higher rates by better following instructions, better using tools, self-correcting previous errors, and verifying candidate solutions.
>     - Even the strongest models are unable to use open-source tools without error, highlighting a persistent limitation in real-world deployment.
>     - On the other hand, our large effect sizes on the importance of verification and self-correction gives deployers reason to think that success on particular benchmarks could also translate to more general problem solving skills.
>     - These insights directly highlight the utility and reliability of AI agents in the wild.
>   - Our experiment design compares different agent scaffolds head-to-head across benchmarks. While previous work has focused on doing similar analysis for a single benchmark (e.g., SWE-Agent on SWE-bench), we conduct this analysis across benchmarks. This allows us to find insights such as task-specific agents outperforming generalist agents, and the drastic impact that scaffolds have across model families (e.g., Anthropic models working better with BrowserUse on Online Mind2Web, and OpenAI models working better with SeeAct).
>   - Our focus on cost also allows us to compare the costs between different scaffolds (e.g., BrowserUse being substantially more expensive than SeeAct).
> - **Novel artifacts**:
>   - A major contribution from HAL is the system-building contribution. It is common for systems papers to solve a series of small challenges in a way that leads to an overall system that is performant and elegant, rather than one big breakthrough insight; Table 2 lists such challenges that we needed to solve for agent evaluation.
>   - As the reviewer notes, we collect and release a 2.5 billion token dataset of 20,000+ agent trajectories to allow researchers to continue our work of exploring, categorizing, and analyzing these results.
>   - This release was only possible given the depth of our engineering effort to build a standardized evaluation harness.
>
> ## 2. Lack of technical depth in the main paper
>
> We thank you for raising this concern. While owing to space limitations we can't discuss the technical details in the main paper, we have significantly updated our discussion in the Appendix (A12) to add technical details about the HAL system architecture. We have also added implementation details, such as about the cost-tracking mechanism, how we implemented adding additional models, etc.

---

> ### Author Response · Authors · 2025-11-24
> **(continued)**
>
> ## 3. Concerns about log analysis
>
> - HAL introduces two novel aspects to the automated log analysis method in addition to the scoring criteria.
> - First, the specific strategy of connecting LLM-as-a-judge flags to task-level success and running correlative analysis is a substantial contribution above and beyond the Docent tool.
> - These details are discussed in depth in Appendix A7, which includes a full description of the methodology, a table of our rubrics and example behaviors, results from human validation, and a large number of screenshots of specific behaviors.
> - Second, manual human validation of the Docent tool has not been done in prior work. We report the results of this validation in Appendix A7.2.
> - The categories used for the log analysis (instruction following, tool use, environmental barriers, self-correction, verification, and benchmark gaming) were created as follows.
> - Log analysis provides an understanding of agents and evaluations across three dimensions: agent capability, agent reliability, and benchmark validity. Instruction following and tool use encompass capability. Self-correction and verification encompass reliability. And environmental barriers and benchmark gaming encompass benchmark validity.
> - Combining these six categories across the three dimensions provides a comprehensive view of both agent behavior and benchmarking.
> - The full process for log analysis is further detailed in Appendix A7.1. We have added this explanation of the rationale behind determining the six categories to this section.
>
> ## 4. Missing artifacts
>
> We apologize for not being able to upload the artifacts alongside the original submission. Owing to the large-scale system design, many aspects of the data can't be uploaded without deanonymizing the authors, which is against ICLR policy during review. That said, in response to your feedback, we have uploaded anonymized versions of 2,000 agent transcripts to this URL: https://anonymous.4open.science/r/hal-paper-analysis-4A24/qualitative/results/transcripts.csv.zip
>
> **We hope this addresses your concerns. If you need additional clarifications or experiments, please let us know**.

---

> ### Comment · Reviewer_YJJ9 · 2025-11-25
>
> I would like to thank the authors for their detailed responses and the additional effort put into the new analyses and data release. I have carefully reviewed the rebuttal and the updated points.
>
> However, my primary concerns regarding the positioning and contribution of this work remain unresolved.
>
> If the primary selling point is **novelty** (methodological or empirical), the rebuttal has not sufficiently addressed the concern that the findings, such as "reasoning effort does not always correlate with accuracy" or the existence of specific failure modes, largely confirm existing intuitions or parallel recent findings in the field rather than offering an interesting pushup. Furthermore, the methodological approach (e.g., the specific log analysis or Pareto frontiers) lacks sufficient comparative analysis with related work to establish it as a novel framework rather than just an application of existing concepts.
>
> If the primary selling point is the **engineering contribution** (as a "system-building" paper), the current submission falls short of the standards expected for an infrastructure paper. To claim this as the main contribution, the work may better offer a "plug-and-play" system that is immediately useful to the community. Currently, there is a lack of documentation, user guides, or a fully accessible, reproducible codebase that would allow other researchers to easily adopt HAL. While I acknowledge the release of the anonymized transcript dataset, this is data, not the infrastructure itself.
>
> The paper currently occupies an ambiguous position. It lacks the deep methodological novelty required for a research paper, yet it lacks the usability, documentation, and open-source completeness required for a pure systems paper. To merit a higher score, the authors would need to decisively commit to one direction: either by providing rigorous comparative experiments to prove unique novelty or by delivering a fully documented, deployable artifact.
>
> As it stands, I am not convinced that the contribution is significant enough in either dimension, and I maintain my original rating.

---

> > ### Author Response · Authors · 2025-11-25
> > **Release of codebase for the infrastructure powering HAL**
> >
> > Thank you for your comment. Over the course of the day, we spent a substantial amount of time to remove personally identifying information from the repository for the paper, and **we have now uploaded the anonymous repository for HAL here**: https://anonymous.4open.science/r/hal-harness-B74E/README.md
> >
> > This includes the entire system, including documentation, a how-to guide to get started, and the entire codebase that would allow researchers to use HAL directly.
> >
> > We were committed to contributing and open-sourcing this infrastructure and adding it to the paper draft once public, but **since the reviewer mentions this is the main drawback in their view from the perspective of a system-building paper, we decided to put in the time to anonymize the entire codebase for the HAL infrastructure and upload it to the link above.**
> >
> > We hope this addresses the reviewer's concern about the lack of release for the infrastructure. If there are still any remaining concerns, **please let us know and we will do our best to address them.**

---

> ### Author Response · Authors · 2025-11-27
> **Requesting an update on the reviewer response**
>
> We thank reviewer YJJ9 for reviewing the paper and engaging with their responses. The reviewer mentioned that the main drawback in their view is the lack of the code/infra for HAL:
>
> > Currently, there is a lack of documentation, user guides, or a fully accessible, reproducible codebase that would allow other researchers to easily adopt HAL.
>
> **We have since added the entire repository for running HAL**, at a significant cost to the author team (since we needed to anonymize every instance of personally identifiable information in an extremely large codebase). The repository can be accessed here: https://anonymous.4open.science/r/hal-harness-B74E/README.md
>
> **Given the short window for engaging with the reviewer, we want to quickly check if the reviewer has any other questions or feedback for us that we might be able to answer or address.** Please let us know.

---

### Official Review · Reviewer_Uqtf · 2025-11-01

**Soundness:** 2
**Presentation:** 4
**Contribution:** 3
**Rating:** 4
**Confidence:** 4

**Summary:**

The authors present an evaluation framework for benchmarking large language models. In doing so, they produce a meaningful evaluation of current models along cost and novel accuracy axes. Among other things, their work finds that a more expensive model is often not more performant than a cheaper model.

**Strengths:**

- Good work. With the number of different benchmarks coming out and the way people are increasingly gaming them, this kind of evaluation goes a long way. And with an inference cost of 40K, this provides a valuable sample for the community.
- The kind of failures mentioned align with what I've experienced in my own work. It's nice to see something that actually disseminates what the key subtle failures of LLMs in agentic works (e.g., I've also seen my agents cheat like this).

**Weaknesses:**

- I feel the contribution of this work is a bit weak, but I think the value of the data point this provides (see Strengths) is enough to outweigh this.
- I have some concerns about the statistical rigour of the results, but I can look past that. What I can't really look past is the analysis based on the Pareto frontier. Most would say that they prefer a model that scores a 9 out of 10 on 100 tasks than any of the 100 other models that each score 10 out of 10 on 1 task but 1 out of 10 on all of the other tasks. With the Pareto frontier, we would say the first model is the worst model by far. This is an even bigger problem, as we know that models are being trained to game the evaluations, so we expect that some models will excel on single tasks but perform poorly overall (i.e., overfitting models will be preferred). I would like to see some kind of smoother metric here. Maybe the average normalized distance to the Pareto frontier? An additional comparison with that is enough.
- For Discovery 2, and related to the above, given how you've drawn Pareto frontiers there using convex hulls, wouldn't that be pretty much guaranteed?

**Questions:**

See Weaknesses.

---

> ### Author Response · Authors · 2025-11-24
> **Additional analysis with an updated metrics, response to concerns**
>
> Thank you for taking the time to review our submission. We appreciate your feedback on our analysis from on the Pareto frontiers. We are glad that you found HAL’s results valuable.
>
> ## 1. Concerns about Pareto Frontier analysis
>
> Thank you for raising this concern. We understand the limitations of binary metrics such as inclusion on the Pareto frontier. Based on your suggestion, we have added model comparisons through a smoother metric (average normalized distance to the Pareto frontier).
>
> | Model                  | AssistantBench | CORE-Bench Hard |  GAIA  | Online Mind2Web | SciCode | ScienceAgentBench | SWE-bench Verified Mini | TauBench Airline |  USACO | Average |
> |------------------------|---------------:|----------------:|-------:|----------------:|--------:|------------------:|------------------------:|-----------------:|-------:|--------:|
> | Gemini 2.0 Flash       |         0.0154 |          0.0128 | 0.0000 |           0.0000|  0.0000 |            0.0000 |                 0.0000  |           0.0000 | 0.0000 |  0.0031 |
> | o4-mini Low            |         0.0000 |          0.0165 | 0.0000 |           0.0366|  0.0000 |            0.0000 |                 0.0174  |           0.0714 | 0.0562 |  0.0220 |
> | o4-mini High           |         0.0565 |          0.0000 | 0.0000 |           0.0410|  0.0068 |            0.0240 |                 0.0228  |           0.0917 | 0.0000 |  0.0270 |
> | DeepSeek V3            |         0.1135 |          0.0000 | 0.0723 |           0.0011|  0.0241 |            0.0195 |                 0.0000  |           0.0574 | 0.0000 |  0.0320 |
> | GPT-4.1                |         0.0668 |          0.0361 | 0.0000 |           0.0000|  0.0226 |            0.0246 |                 0.0612  |           0.0920 | 0.0146 |  0.0353 |
> | o3 Medium              |         0.0000 |          0.0662 | 0.0981 |           0.0030|  0.0000 |            0.0000 |                 0.0864  |           0.1581 | 0.0690 |  0.0534 |
> | GPT-5 Medium           |         0.1709 |          0.0128 | 0.0000 |           0.0000|  0.1168 |            0.0870 |                 0.0000  |           0.1339 | 0.0000 |  0.0579 |
> | Claude-3.7 Sonnet High |         0.0918 |          0.0139 | 0.0748 |           0.1235|  0.1292 |            0.0130 |                 0.0000  |           0.0235 | 0.1404 |  0.0678 |
> | Claude-3.7 Sonnet      |         0.2361 |          0.0000 | 0.1363 |           0.0733|  0.1174 |            0.0322 |                 0.0061  |           0.0000 | 0.0996 |  0.0779 |
> | DeepSeek R1            |         0.3154 |          0.0541 | 0.1400 |           0.0464|  0.1936 |            0.0963 |                 0.0896  |           0.1636 | 0.1331 |  0.1369 |
> | Claude Opus 4.1        |         0.5583 |          0.0000 | 0.3381 |              —  |  0.3608 |            0.2041 |                 0.0250  |           0.3232 | 0.2892 |  0.2623 |
> | Claude Opus 4.1 High   |         0.6704 |          0.0414 | 0.3691 |              —  |  0.3502 |            0.2068 |                 0.0000  |           0.2940 | 0.2793 |  0.2764 |
>
> This table relies on calculating the frontier and distances from the frontier using the log of cost. We made this decision for three reasons.
> 1. First, as the frontier includes the origin, we felt that linear cost metrics disproportionately advantaged cheap models that also had low accuracy.
> 2. Second, the range of cost on some of our benchmarks can be quite large, up to three orders of magnitude.
> 3. Third, we perceive most consumers as principally optimizing for accuracy with a constraint of cost. Thus, measuring our cost in log terms gives an interpretation to our distance that more closely conforms to “what multiplicative change in cost would I exchange for a percentage increase in accuracy.”
>
> ## 2. Concern about only including points on the convex hull of the Pareto frontier
>
> - We agree that using the convex hull makes it less likely for models to appear on the frontier. Although it is more likely that the Pareto frontier of accuracy and cost would be sparse, our results still confirm that this is true in practice, since it is theoretically possible for many more models to be on the convex hull, such as if models have high differentiation, few internal model routers, and diminishing returns would have this feature.
> - **We will reword this finding in the paper to add that using the convex hull makes this more likely.**
>
> Once again, we thank the reviewer for taking the time to review the paper. **Please let us know if you need additional details, clarifications, or experiments.**

---

> > ### Author Response · Authors · 2025-11-27
> > **Requesting an update on the reviewer response**
> >
> > We thank reviewer Uqtf for reviewing the paper. The reviewer mentioned that the main drawback in their view is the lack of an aggregate measure:
> > > I would like to see some kind of smoother metric here. Maybe the average normalized distance to the Pareto frontier? An additional comparison with that is enough.
> >
> > **We have since added this smoother metric** (specifically, the metric the reviewer suggested, average normalized distance to the Pareto frontier). It is included in the response to the reviewer above, as well as in the updated PDF for the paper.
> >
> > Given the short window for engaging with the reviewer, **we want to quickly check if the reviewer has any other questions or feedback for us that we might be able to answer or address.** Please let us know.

---

### Official Review · Reviewer_AkiS · 2025-11-08

**Soundness:** 3
**Presentation:** 3
**Contribution:** 2
**Rating:** 6
**Confidence:** 3

**Summary:**

I was invited as a supplementary reviewer for this paper. Since I received it less than five working days before the ICLR 2026 deadline on November 11, I may have missed some details while reading. If anything in my understanding is incorrect, I would appreciate it if the authors could point it out.

This paper tries to tackle several long-standing challenges in evaluating AI agents. The authors argue that current evaluations often suffer from inconsistent setups, high cost, long evaluation cycles, and sometimes untrustworthy results. To address these issues, they introduce the Holistic Agent Leaderboard, or HAL, which combines a unified evaluation infrastructure, a three-dimensional leaderboard across models, scaffolds, and benchmarks, and an automated log analysis system.

Through this framework, the authors conduct large-scale experiments and report a number of interesting findings. For example, they observe that higher reasoning effort does not always lead to better accuracy, and that the relationship between cost and performance is often nonlinear.

**Strengths:**

- [Large-scale and carefully executed empirical study] Running over 21,000 rollouts across 9 models and 9 benchmarks is nontrivial, both financially and logistically. The authors’ effort to document pricing, token usage, and performance variance across models gives the paper a solid quantitative backbone. The Pareto analysis between cost and performance is convincing and sets a precedent for cost-aware evaluation in future agent work.

- [Balanced benchmark and leaderboard design with meaningful difficulty] The authors have managed to calibrate the benchmark difficulty very well. Across the nine included tasks, there is enough separation among different agents to expose real capability differences, while state-of-the-art systems still leave considerable room for improvement. This balance suggests that the benchmark is neither trivial nor saturated, which makes it more likely to remain useful and relevant as agent technology evolves. The diversity of domains (coding, science, web navigation, and customer service) further helps prevent overfitting to a single narrow task type.

- [Strong systems thinking and reproducibility awareness]  The authors manage to unify heterogeneous agent scaffolds, benchmarks, and execution environments through a minimal API interface. The orchestration across hundreds of virtual machines, coupled with built-in token accounting and cost normalization, reflects an impressive level of engineering maturity.

**Weaknesses:**

- I acknowledge the strong engineering effort demonstrated in this work, but the paper feels overly engineering-oriented. It reads more like a large-scale system report than a research study that produces new principles or algorithms. The authors repeatedly emphasize that HAL “standardizes evaluation” and “makes agent benchmarking reproducible,” yet the methodological novelty remains quite thin. The reported 21,000 rollouts are impressive in scale, but the results mostly confirm intuitive trends: expensive models cost more, scaffolds matter, and reasoning helps inconsistently. The large experimental effort does not lead to equally deep conceptual insight. In several figures, the analysis stops at “we observe X,” without further investigation into why the phenomenon occurs or what it implies for the design of future agents. This is the main reason I give the paper a score of 6 instead of 8.

- Another concern is that several parts of the paper feel ad hoc in both design and analysis. The selection of benchmarks lacks a clear organizing principle. The nine included tasks span very different domains but do not follow a consistent taxonomy or rationale for why these particular tasks define a “holistic” view of agent capabilities. Moreover, the behavioral log analysis uses author-defined categories without theoretical grounding or validation, leading to largely descriptive findings. Overall, the paper reads as a collection of interesting observations rather than a systematically constructed methodology, which weakens its claim of being a “standardized evaluation framework.”

- If I did not miss it, the paper does not include any real comparison between HAL and existing evaluation frameworks. I am not referring to the brief listing in Appendix A1, but to a more meaningful validation such as consistency with human expert judgments or correlation with established benchmarks. Without such analysis, there is no quantitative evidence showing that HAL actually provides a more reliable or more accurate evaluation than existing methods. As a result, the claim that HAL improves standardization and trustworthiness remains largely unsubstantiated.

**Questions:**

I see this work mainly as a benchmark and large-scale analysis paper. I do not have particular questions for the authors at this stage. My earlier comments already cover the key concerns I have about framing, validation, and the depth of insight.

---

> ### Author Response · Authors · 2025-11-24
> **Responses to reviewer concerns, additional analysis**
>
> Thank you for taking the time to review our submission. We appreciate your feedback on providing more details about the key contributions of our work outside of the engineering effort and the methodological novelty. We are glad that you found HAL balanced and comprehensively designed, and thoughtfully engineered.
>
> ## 1. Concern that the paper is overly engineering-oriented
>
> Thanks for raising this concern. We appreciate the perspective, and can understand how our contributions were not spelled out enough in the paper draft. We have updated the paper's text to reflect the contributions we have made to the analytical methods and insights, and also added results that we don't think just confirm intuitive findings but rather are a step forward for the community.
>
> - **Novel insights**:
>   - Many of the results we found were first reported in HAL, and couldn't have been reported without the large-scale system we built.
>   - For example, we comprehensively evaluate the effect of low vs. high reasoning across benchmarks, and find that higher reasoning does not help performance in the majority of cases. Given the strong claims about the impact of reasoning effort made by companies (such as GPT-5's results on Charxiv and SWE-Bench, and o3 and o4-mini's results on GPQA and AIME), we think it is important to test these claims empirically. Otherwise there is too much dependence on ''folk wisdom'' that isn't actually true and leads the community astray.
>   - We also found evidence of reliability failures through log analysis, such as agents refunding excessive amounts, charging the wrong credit card etc.
>   - We observe that increases in accuracy are not accompanied by decreases in token usage, on average. It is significant that stronger models are not displaying more efficient planning and execution.
>   - Our log analysis reveals new insights, including.
>     - the dramatic reliability failures in TauBench which would be catastrophic in deployment,
>     - detecting a bug in the official implementation of a popular benchmark's scaffold,
>     - stronger models succeed at higher rates by better following instructions, better using tools, self-correcting previous errors, and verifying candidate solutions.
>     - Even the strongest models are unable to use open-source tools without error, highlighting a persistent limitation in real-world deployment.
>     - On the other hand, our large effect sizes on the importance of verification and self-correction gives deployers reason to think that success on particular benchmarks could also translate to more general problem solving skills.
>     - These insights directly highlight the utility and reliability of AI agents in the wild.
>   - Our experiment design compares different agent scaffolds head-to-head across benchmarks. While previous work has focused on doing similar analysis for a single benchmark (e.g., SWE-Agent on SWE-bench), we conduct this analysis across benchmarks. This allows us to find insights such as task-specific agents outperforming generalist agents, and the drastic impact that scaffolds have across model families (e.g., Anthropic models working better with BrowserUse on Online Mind2Web, and OpenAI models working better with SeeAct).
>   - Our focus on cost also allows us to compare the costs between different scaffolds (e.g., BrowserUse being substantially more expensive than SeeAct).
> - **Novel artifacts**:
>   - A major contribution from HAL is the system-building contribution. It is common for systems papers to solve a series of small challenges in a way that leads to an overall system that is performant and elegant, rather than one big breakthrough insight; Table 2 lists such challenges that we needed to solve for agent evaluation.
>   - As the reviewer notes, we collect and release a 2.5 billion token dataset of 20,000+ agent trajectories to allow researchers to continue our work of exploring, categorizing, and analyzing these results.
>   - This release was only possible given the depth of our engineering effort to build a standardized evaluation harness.

---

> > ### Author Response · Authors · 2025-11-24
> > **(continued)**
> >
> > ## 2. Further investigation of the results
> > - We would like to note that the paper does include quantitative analysis of common failure modes and reliability correlates; this already helps readers dig deeper into the causes of agent failures and how agents can be improved.
> > - But based on this feedback, we’ve deepened our analytical approach by building a logistic regression model which fits the probability of task success as a function of the benchmark, task, model, length of the trajectory, and our five behavioral flags.
> > - When controlling for these other variables, we observe the following odds ratios for each of our flags:
> >
> > | *Behavior* | *Odds Ratio (multiplicative effect)* |
> > | ----- | ----- |
> > | *Self-Correction* | *1.58* |
> > | *Verification* | *1.87* |
> > | *Tool Calling Failure* | *0.57* |
> > | *Instruction Following Failure* | *0.31* |
> > | *Environmental Barrier* | *0.54* |
> >
> > - Each of these results has the expected sign, is statistically significant, and has a 95% confidence interval that includes either a doubling or a halving of the probability of success. **We will add the results of this regression-based approach to log analysis to the paper.**
> >
> > ## 3. The section of benchmarks lacks a clear organizing principle
> >
> > - Thanks for raising this concern. We are adding more details about our benchmark selection methodology to the paper.
> > - To select benchmarks for integration with HAL, we identified five major domains where agents are being developed and deployed: web navigation, software engineering, scientific research, customer service, and cybersecurity.
> > - We then surveyed 30 agent benchmarks spanning these five domains identified by Kapoor et. al [4], Zhu et. al [5], and our own prior knowledge as benchmarks used by researchers and AI providers to measure agent capabilities. **The benchmarks are listed in Appendix A10 in the updated PDF for the paper**.
> > - From these 30 benchmarks, we selected 1-3 benchmarks from each of the 5 domains that represent tasks with high construct validity.
> > - As mentioned in the paper, we left the addition of a cybersecurity benchmark to future iterations of HAL.
> > - The 9 benchmarks we selected are listed in Table 3. This selection ensures breadth and coverage of typical agent applications. They test whether agents can navigate complex interfaces, write correct code, conduct scientific analysis, and handle multi-turn interactions.
> > - Note that the list of domains is not exhaustive; there are many other domains that we do not focus on for this version of HAL, such as retail. While this is outside the scope of the current paper, we plan to increase the scope of our evaluations in future iterations of HAL.
> > - We have edited Section 3 with this explanation of our benchmark selection process, taxonomy, and rationale.
> >
> > ## 4. Concern about log analysis methodology
> > - Since past work has not thoroughly conducted such evaluations at a large scale, and best practices are still evolving, we develop our rubrics for agent transcript analysis by expanding on the best available evidence from prior work (such as Cemri et al. [6]).  We hope that our work can form the basis for future attempts at automated log analysis.
> > - We validated the rubric we used for log analysis against manual validators, and have included the results in the table above (we found that all five categories of log analysis yield significant results).
> > - The full process for log analysis is further detailed in Appendix A7.1. We have added an explanation of the rationale for these six categories to this section.

---

> > > ### Author Response · Authors · 2025-11-24
> > > **(continued)**
> > >
> > > ## 5. Comparison between HAL and existing evaluation frameworks
> > > -  It is difficult to compare HAL to other existing evaluation frameworks since there are no other existing multi-domain, standardized evaluation platforms that serve the same purpose as HAL.
> > > - That said, we do perform manual validation of the log analysis results, by comparing the characterization of failure modes with human judgement. We report the results of this validation in Appendix A7.2.
> > > - Additionally, **we have added empirical analysis to the paper** showing how similarly different benchmarks rank models. While this does not help readers compare HAL to external evaluations, it provides a clearer sense of how consistent performance is across benchmarks in HAL.
> > > - We compute pairwise benchmark rank correlation of model accuracy (using the scaffold with highest accuracy). We then calculate the mean of these rank correlations to get a single correlative statistic of 0.46. We observe even higher rank correlations for related benchmarks, e.g. 0.58 for scientific programming and 0.58 for web assistance. (The updated paper includes a heatmap, we are adding a table below for convenience)
> > >
> > > | Benchmark                | AssistantBench | CORE-Bench Hard |   GAIA   | Online Mind2Web | SWE-bench Verified Mini |  SciCode  | ScienceAgentBench |   USACO   |
> > > |--------------------------|---------------:|----------------:|---------:|----------------:|------------------------:|----------:|------------------:|----------:|
> > > | AssistantBench           |      1.000000  |        0.066667 | 0.721542 |         0.466667|                0.250515 | 0.545175  |          0.691385 | 0.412587  |
> > > | CORE-Bench Hard          |      0.066667  |        1.000000 | 0.244289 |         0.682940|                0.768552 | 0.577287  |          0.371687 | 0.315791  |
> > > | GAIA                     |      0.721542  |        0.244289 | 1.000000 |         0.490909|                0.265294 | 0.242922  |          0.498246 | 0.367776  |
> > > | Online Mind2Web          |      0.466667  |        0.682940 | 0.490909 |         1.000000|                0.435616 | 0.635912  |          0.628060 | 0.248485  |
> > > | SWE-bench Verified Mini  |      0.250515  |        0.768552 | 0.265294 |         0.435616|                1.000000 | 0.663083  |          0.519910 | 0.093048  |
> > > | SciCode                  |      0.545175  |        0.577287 | 0.242922 |         0.635912|                0.663083 | 1.000000  |          0.801791 | 0.509774  |
> > > | ScienceAgentBench        |      0.691385  |        0.371687 | 0.498246 |         0.628060|                0.519910 | 0.801791  |          1.000000 | 0.391550  |
> > > | USACO                    |      0.412587  |        0.315791 | 0.367776 |         0.248485|                0.093048 | 0.509774  |          0.391550 | 1.000000  |
> > >
> > > Once again, we thank the reviewer for taking the time to review the paper. **Please let us know if you need additional details, clarifications, or experiments.**
> > >
> > > [1] Liang et al. Holistic Evaluation of Language Models. TMLR (2023).
> > >
> > > [2] Chiang et al. Chatbot Arena: An Open Platform for Evaluating LLMs by Human Preference. ICML (2024).
> > >
> > > [3] Gao et al. A framework for Few-shot Language Model Evaluation (2021).
> > >
> > > [4] Kapoor et. al. AI Agents That Matter. TMLR (2025).
> > >
> > > [5] Zhu et al. Establishing Best Practices for Building Rigorous Agentic Benchmarks. NeurIPS D&B (2025).
> > >
> > > [6] Cemri et al. Why Do Multi-Agent LLM Systems Fail? (2025).

---

> > > > ### Comment · Reviewer_AkiS · 2025-11-25
> > > >
> > > > Thank you for the detailed response. I have read the other reviewers’ comments as well as the authors’ rebuttal. Overall, I still tend to maintain my original score of 6. In addition, I noticed that almost all reviewers pointed out that the paper is overly engineering-oriented, which has also raised concerns about its novelty.
> > > >
> > > > I appreciate the authors’ argument that “many of the insights we report would not be possible without such engineering effort, leading to folk wisdom in the community that isn't actually true.” However, I would like to gently remind the authors that, for ICLR submissions, readers typically expect a paper to solve, propose, or at least discuss a scientific question, rather than primarily an engineering one. Technical details can be shortened or moved to the appendix (as is often required by journals such as Nature).
> > > >
> > > > It may help to reconsider the work from the perspective of its intended audience. For a benchmark-style paper, potential readers might include newcomers to the agent field, companies seeking agent-based solutions, or agent developers. Their interests are diverse, but they are unlikely to be aiming to build yet another Agent Leaderboard, and even less likely to care about low-level logs and similar engineering artifacts—these are typically relevant only to a small subset of close technical peers.
> > > >
> > > > I hope the authors can take this into account and ensure that valuable scientific insights are not overshadowed by engineering details.

---

> ### Author Response · Authors · 2025-11-27
> **Scope for ICLR, insights for various stakeholders**
>
> Thank you for your quick response, we appreciate the engagement with the paper. We have a few brief responses to the reviewer's concerns:
> >I would like to gently remind the authors that, for ICLR submissions, readers typically expect a paper to solve, propose, or at least discuss a scientific question, rather than primarily an engineering one. Technical details can be shortened or moved to the appendix (as is often required by journals such as Nature).
> - **Scope for ICLR**: We want to note that [ICLR's call for papers](https://iclr.cc/Conferences/2026/CallForPapers) has **two specific line items** on the exact type of contribution we make: **1) datasets and benchmarks, 2) infrastructure, software libraries, hardware, etc.** Given this focus, and the strong list of papers in past versions of ICLR that make such contributions and have gone on to be extremely impactful, as well as the presence of novel insights in our paper, we think our paper fits the ICLR call for papers very well. In particular, we release a dataset consisting of 2.5B tokens of LLM calls for solving agentic tasks across 9 benchmarks and models, and we release the infrastructure for conducting agent evaluation at scale.
>
> > For a benchmark-style paper, potential readers might include newcomers to the agent field, companies seeking agent-based solutions, or agent developers. Their interests are diverse, but they are unlikely to be aiming to build yet another Agent Leaderboard, and even less likely to care about low-level logs and similar engineering artifacts—these are typically relevant only to a small subset of close technical peers.
>
> We agree that the set of stakeholders you identify are unlikely to want to build an agent leaderboard or care about engineering artifacts. But that's not what HAL offers them! In particular, here are just some examples of the insights these different stakeholders can get from HAL:
> - **Companies seeking agent-based solutions** can identify potential failure modes that would be very costly in deployment. For example, last year, [Air Canada was made to refund a customer](https://www.bbc.com/travel/article/20240222-air-canada-chatbot-misinformation-what-travellers-should-know) based on flawed information from their chatbot. We found similar errors (that could be even more costly) from benchmarks that represent real-world applications like customer service; for example on Tau-Bench, even leading models can refund users unnecessarily, book flights to and from the incorrect destination etc. We think companies seeking agent-based solutions could benefit greatly from understanding these failure modes from real-world settings, and our results could also prevent overreliance on benchmark accuracy numbers alone, as they hide many real-world failures, showing the need for log analysis.
> - **Agent developers conducting agent evaluation on benchmarks.** HAL makes it easy for agent developers to evaluate their  agents on existing benchmarks and easily add new benchmarks. They can get features such as cost logging, log analysis, and Pareto frontiers from the infra. (Since we released HAL, two major companies developing AI agents are using HAL, and two leading AI model developers are supporting the effort with API credits.)
> - **Agent developers can identify patterns in agent design that help across benchmarks.** For example, we found behaviors such as self-verification and self-correction are correlated with stronger agent performance across benchmarks; these behaviors could be incentivized during agent design to improve agent accuracy.
> - **Agent developers can identify the benefits of task-specific agents.** Many agent developers are faced with an existential question: can their agents continue to provide value over and above just using the latest model equipped with a set of tools (such as on popular interfaces like ChatGPT)? We show that generalist agents continue to lag behind task-specific agents across benchmarks. This provides clear evidence that making task-specific modifications can have drastic impact on the agent's overall accuracy.
> - **Newcomers to the agents field can avoid being misled by folk wisdom.** We identify many cases where our comprehensive results differ from commonly accepted folk wisdom in the community. We think a newcomer to the field would be well suited to learning from the **results** and **insights** of our paper, rather than merely the engineering effort.
> - **Engineering details vs. scientific insights.** We appreciate the reviewer's suggestion that we could include fewer engineering details to focus more on the scientific insights. However, we aimed to strike a balance between introducing details of engineering, details on insights, and details on novel methods (such as large-scale log analysis for benchmark logs). As you might have noticed, another reviewers requested us to add *more* details of the implementation to the paper; we aimed to strike a balance between these two.

---

> > ### Comment · Reviewer_AkiS · 2025-11-27
> >
> > Considering what has just happened, I think it is better for me to respond as soon as possible. Since I am a “last-minute” reviewer, and my research area is not closely aligned with yours, there are some details that I am unable to fully grasp. However, I believe that I have still provided a relatively fair, detailed, and responsible evaluation within the limits of my expertise. I remain generally positive about the paper, and I trust that the AC will take this into account during the decision process. Thank you again for your response, and I hope you can understand and support my decision.
> >
> > As a personal note, under the current circumstances, I believe that the numerical score may no longer carry much weight, and the AC and PC will likely rely primarily on the original reviews and the rebuttal to make the final decision.

---

### Official Review · Reviewer_nPVd · 2025-11-11

**Soundness:** 3
**Presentation:** 3
**Contribution:** 4
**Rating:** 8
**Confidence:** 3

**Summary:**

The paper introduces the Holistic Agent Leaderboard (HAL), a unified evaluation framework for AI agents. HAL promotes standardized engineering practices for evaluation, providing a harness that features cloud-based orchestration for parallel execution and environment management, unified logging (via Weave), and automated log analysis (via Docent). The framework is structured to enable an orthogonal analysis across three dimensions: models, scaffolds, and benchmarks.

To validate this framework, the authors conducted 21,730 agent rollouts, testing 9 models across 9 benchmarks in domains including coding, web navigation, scientific research, and customer service. The analysis reports both accuracy and cost, computing the Pareto frontier for these two metrics.

The automated log analysis identified specific agent behaviors, such as taking shortcuts by finding benchmark answers on HuggingFace or making operational errors like misusing a credit card for a flight booking. The analysis also identified flaws in evaluation setups, including data leakage in an official scaffold. The authors have publicly released the open-source HAL harness, the leaderboard, and all 2.5 billion tokens of collected agent logs.

**Strengths:**

The paper presents a thoughtfully engineered system that addresses practical challenges in agent evaluation. It uses cloud-based orchestration for parallel execution, standardizes setups by decoupling scaffold from benchmark execution, and automates logging and trace analysis. The engineering work itself makes it a useful tool for agent researchers and practitioners.

The experiment design is comprehensive (covering 9 models across 9 benchmarks in 4 domains and totaling 21,730 rollouts). This relatively large-scale study helps drive convincing results on cost-accuracy tradeoffs via Pareto analysis, and identifies interesting reward hack patterns like searching on HuggingFace.

The authors have released the open-source HAL harness, the public leaderboard, and the complete agent logs of  21,730 rollouts, providing a resource for research on agent behavior and reliability.

**Weaknesses:**

Although the paper provides a useful engineering tool and a large-scale evaluation, its primary analytical methods and insights are not particularly novel. For instance, the cost-accuracy Pareto frontier analysis was previously established as a core evaluation principle by prior work (e.g. https://arxiv.org/abs/2407.01502).

Similarly, the reward hacking behavior identified, such as searching for benchmark answers, is a concrete example of the known problem as discussed in Search-Time Data Contamination (https://arxiv.org/abs/2508.13180). Given the 2.5 billion token trajectory dataset, it would be great if the authors could offer more unique methods/insights beyond these known issues, such as a systematic clustering of common failure patterns or a quantitative analysis of inefficient agent trajectories, etc.

**Questions:**

The paper's reasoning effort analysis focuses on model-level API parameters. However, complex scaffolds may have their own reasoning budget/context parameters (e.g., max_observation_length for swe-agent). Could the authors provide more details here, and assess if there's any impact on the evaluation results?

---

> ### Author Response · Authors · 2025-11-24
> **Additional analysis and responses to concerns about novelty**
>
> Thank you for your review. We appreciate the engagement with our paper. We especially appreciate the reviewer considering our system thoughtfully engineered and our experiment design comprehensive.
>
> We address your questions and concerns below, and have **conducted additional empirical analysis** in response to your concerns. We hope this addresses your concerns, but **please let us know if you need additional details, experiments, or clarifications.**
>
> ## 1. Concern around novelty
>
> We appreciate the concern about the novelty in the paper. That said, we do think there are several areas where the paper contributes novel methods, insights, and artifacts. We list some of these below, and have updated the paper PDF to clarify these contributions:
>
>
> - **Novel insights**:
>   - Many of the results we found were first reported in HAL, and couldn't have been reported without the large-scale system we built.
>   - For example, we comprehensively evaluate the effect of low vs. high reasoning across benchmarks, and find that higher reasoning does not help performance in the majority of cases. Given the strong claims about the impact of reasoning effort made by companies (such as GPT-5's results on Charxiv and SWE-Bench, and o3 and o4-mini's results on GPQA and AIME), we think it is important to test these claims empirically. Otherwise there is too much dependence on ''folk wisdom'' that isn't actually true and leads the community astray.
>   - We also found evidence of reliability failures through log analysis, such as agents refunding excessive amounts, charging the wrong credit card etc.
>   - We observe that increases in accuracy are not accompanied by decreases in token usage, on average. It is significant that stronger models are not displaying more efficient planning and execution.
>   - Our log analysis reveals new insights, including.
>     - the dramatic reliability failures in TauBench which would be catastrophic in deployment,
>     - detecting a bug in the official implementation of a popular benchmark's scaffold,
>     - stronger models succeed at higher rates by better following instructions, better using tools, self-correcting previous errors, and verifying candidate solutions.
>     - Even the strongest models are unable to use open-source tools without error, highlighting a persistent limitation in real-world deployment.
>     - On the other hand, our large effect sizes on the importance of verification and self-correction gives deployers reason to think that success on particular benchmarks could also translate to more general problem solving skills.
>     - These insights directly highlight the utility and reliability of AI agents in the wild.
>   - Our experiment design compares different agent scaffolds head-to-head across benchmarks. While previous work has focused on doing similar analysis for a single benchmark (e.g., SWE-Agent on SWE-bench), we conduct this analysis across benchmarks. This allows us to find insights such as task-specific agents outperforming generalist agents, and the drastic impact that scaffolds have across model families (e.g., Anthropic models working better with BrowserUse on Online Mind2Web, and OpenAI models working better with SeeAct).
>   - Our focus on cost also allows us to compare the costs between different scaffolds (e.g., BrowserUse being substantially more expensive than SeeAct).
> - **Novel artifacts**:
>   - A major contribution from HAL is the system-building contribution. It is common for systems papers to solve a series of small challenges in a way that leads to an overall system that is performant and elegant, rather than one big breakthrough insight; Table 2 lists such challenges that we needed to solve for agent evaluation.
>   - As the reviewer notes, we collect and release a 2.5 billion token dataset of 20,000+ agent trajectories to allow researchers to continue our work of exploring, categorizing, and analyzing these results.
>   - This release was only possible given the depth of our engineering effort to build a standardized evaluation harness.

---

> > ### Author Response · Authors · 2025-11-24
> > **(continued response to reviewer nPVd)**
> >
> > ## 2. Additional log analysis experiments
> >
> > - We appreciate the concern about the methodology for the log analysis. We clarify our existing approach and some additional analysis we conducted based on this feedback below.
> > - **Our approach is geared towards clustering common failure modes** rather than relying on ad hoc labeling of individual failures. This process is detailed in Appendix A7.1.
> > - Moreover, **the behaviors identified through log analysis extend well beyond previously known behaviors such as reward hacking**. Our methodology uncovers previously undetected failures in benchmark design and agent behavior, including specific tool use errors, environmental barriers that prevent task completion, and conflicting instructions that create inherent difficulty in solving tasks, which shows the limitations of benchmarks for making conclusions about the real-world capabilities of agents, common agent failure modes, and reliability issues.
> > - **Additional analysis conducted in response to the reviewer's concerns**: in response to this concern, we deepened our analytical approach by building a logistic regression model which fits the probability of task success as a function of the benchmark, task, model, length of the trajectory, and our five behavioral flags. When controlling for these other variables, we observe the following odds ratios for each of our flags:
> >
> > | *Behavior* | *Odds Ratio (multiplicative effect)* |
> > | ----- | ----- |
> > | *Self-Correction* | *1.58* |
> > | *Verification* | *1.87* |
> > | *Tool Calling Failure* | *0.57* |
> > | *Instruction Following Failure* | *0.31* |
> > | *Environmental Barrier* | *0.54* |
> >
> >
> > Each of these results has the expected sign, is statistically significant, and has a 95% confidence interval that includes either a doubling or a halving of the probability of success. **We have added the results of this regression-based approach to log analysis to the paper.**
> >
> > ## 3. Question on agent parameters
> >
> > - While our current analysis does not give us a precise method for analyzing the effect of specific reasoning budget parameters of our agent scaffolds, we used log analysis to analyze the effect of more steps.
> > - Using the same logistic regression causal inference strategy described above, when controlling for benchmark, model, task, and behavioral flags, we observe a very minor positive effect of additional agent steps.
> > - **Results**: We find that each additional five turns increases the probability of task success by only 3%. **We plan to include this additional analysis in the paper**
> >
> > Once again, we thank the reviewer for taking the time to review the paper. **Please let us know if you need additional details, clarifications, or experiments.**

---

> > > ### Comment · Reviewer_nPVd · 2025-11-27
> > >
> > > Thank your thoughtful response and clarification.
> > >
> > > While I do echo with the other reviewers that the positioning of the paper and the presentation of its contributions can be improved, I think the paper offers a good mixture of service and novel findings derived from the engineering work (as reflected in my original scores).
> > >
> > > I will keep my current score.

---

### Official Review · Reviewer_n4gN · 2025-11-11

**Soundness:** 2
**Presentation:** 1
**Contribution:** 4
**Rating:** 6
**Confidence:** 1

**Summary:**

I have read this paper on ArXiv, and therefore know who the authors are. Thus, I am unable to review the paper. I have informed the ACs of this. Please ignore this review. The following options are randomly chosen to avoid biasing other reviewers.

**Strengths:**

Not applicable

**Weaknesses:**

Not applicable

**Questions:**

Not applicable

---

### Author Response · Authors · 2025-11-24
**Overall response, additional experiments, updates to the paper draft**

We thank the reviewers for their comments. We have made significant updates to the paper based on the feedback. While we have responded to their concerns individually, we want to add some high-level aggregate responses here:

## Additional empirical analysis
  - Some of the reviewers' concerns revolved around additional empirical analysis. We have conducted additional analysis on the factors underlying agent successes and failures, the correlations of HAL results across benchmarks, and the impact of additional turns on agent success probability.

## Data release
  -   Another reviewer concern was about the unavailability of the dataset of agent rollouts. While we are unable to share the full dataset at this stage owing to ICLR's anonymization policy (given HAL is a complex project, it is hard to ensure complete anonymity on the large dataset of logs accompanying it), **we have now added an anonymized dataset with 2,000 agent transcripts on this link: https://anonymous.4open.science/r/hal-paper-analysis-4A24/qualitative/results/transcripts.csv.zip**. This link also includes the code to conduct the analysis for the paper's results. We commit to ensuring all data is released publicly once the paper is de-anonymized.

## Concern around novelty
  - We understand the reviewers' concerns about novelty. That said, we would like to point out that a major contribution of HAL is the system-building aspect. It is common for systems papers to solve a series of small challenges in a way that leads to an overall system that is performant and elegant, rather than one big novel insight; Table 2 lists such challenges that we needed to solve for agent evaluation.
  - In addition, many prior works published at top conferences (including ICLR) have contributions that are primarily associated with introducing a new dataset, benchmark, or other artifact, such as SWE-Bench [1], MLE-Bench [2], PRISM [3], RE-Bench [4], LiveBench [5], and have gone on to be extremely impactful.
  - Finally, the experimental setup of HAL does allow us to uncover novel insights about the benchmarks that we evaluate, which we go into further detail below.
  - A few examples:
    - the dramatic reliability failures in TauBench, which would be catastrophic in deployment
    - detecting a bug in the official implementation of a popular benchmark's scaffold
    - finding that higher reasoning effort doesn't increase accuracy in the majority of cases, etc.

##  Concern around engineering focus:
  - It is true that the paper involves a significant engineering effort to conduct 20,000+ agent rollouts. At the same time, many of the insights we report **would not be possible without such engineering effort, leading to folk wisdom in the community that isn't actually true**.
  - For example, companies naturally highlight positive results when announcing new models. OpenAI's announcements for o3 and o4-mini emphasize improvements with higher reasoning effort on benchmarks like AIME and GPQA, while GPT-5's release highlighted gains on SWE-Bench and Charxiv
  - However, our evaluation reveals that higher reasoning effort can actually decrease accuracy by as much as 20 percentage points on other benchmarks for models like o4-mini, Claude Opus, and Claude Sonnet
  - Without systematic evaluation, these negative results receive little attention, creating a distorted view of reasoning as near-universally beneficial. Incorrect beliefs become accepted despite being true only for some benchmark-model combinations.
  - We hope HAL can dispel similar incorrect beliefs in the community.
  - In addition, past work that has carried out similar engineering-heavy contributions (such as HELM [6], Chatbot Arena [7], AgentBoard [8]) has gone on to be extremely impactful, winning a best paper award, and leading to significant downstream work.
  - Finally, we release a dataset with 2.5B tokens of LLM calls. We expect that this artifact would be independently useful as a dataset for spurring future research.

**We welcome any additional questions, requests for clarifications, or analyses that would aid the review process. We look forward to engaging with the reviewers.**

[1] Jimenez et al. SWE-Bench: Can Language Models Resolve Real-World GitHub Issues. ICLR (2024).

[2] Chan et al. MLE-bench: Evaluating Machine Learning Agents on Machine Learning Engineering. ICLR (2025).

[3] Kirk et al. The PRISM Alignment Dataset. NeurIPS (2024).

[4] Wijk et al. RE-Bench: Evaluating Frontier AI R&D Capabilities of Language Model Agents against Human Experts. ICML (2025).

[5] White et. al. LiveBench: A Challenging, Contamination-Limited LLM Benchmark. ICLR (2025).

[6] Liang et al. Holistic Evaluation of Language Models. TMLR (2023).

[7] Chiang et al. Chatbot Arena: An Open Platform for Evaluating LLMs by Human Preference. ICML (2024).

[8] Ma et al. AgentBoard: An Analytical Evaluation Board of Multi-turn LLM Agents. NeurIPS (2025).

---

### Meta-Review · Area_Chair_FjVj · 2026-01-11

**Summary:**

Reviewer n4gN
This is an empty review, I will discount it.

Reviewer nPVd (score 8)
commented that the paper provides a good engineering tool but does not add much in terms of analytical methods or insights.

Reviewer AkiS (score 6)
commented that this manuscript does a good job of pointing to some widely believed trends (reasoning being inconsistent in these tasks) but does not do a good job of saying why it might be the case. The benchmark seems too broad and ad hoc. The reviewer also wanted to see comparisons of the benchmark against other benchmarks, human judgements etc.

Reviewer Uqtf (score 4)
commented on the contributions being weak. They also had some comments on modifying the Pareto frontier metric to be more sound.

Reviewer YJJ9 (score 2)
commented that this is primarily an engineering paper and the analysis is not methodologically novel, e.g., the log analysis tool uses Docent which is an existing system and does not provide enough details of how the analysis was done.

This paper builds a Holistic Agent Leaderboard (HAL) to standardize evaluation of AI agents across 9 models and 9 kinds of benchmarks with more than 20,000 rollouts. The paper analyzes a large number of agent logs, LLM calls (2.5 billion) and has made all the code and data public. Using the data they have shed light on a number questions such as, does higher reasoning effort improve performance, and identified unreported behaviors. While this is a primarily engineering effort, and some of the reviewers were hesitant to recommend an accept, we must recognize that such large-scale data analysis is critical to guide the research on agentic systems. This analysis and the curation of data is as important, if not more, as that of any methodological improvement in this nascent field.

**Reviewer Concerns:**

Reviewer n4gN
NA

Reviewer nPVd
The authors have provided an adequate rebuttal to argue that many of the claims/insights in the paper were either non-existent before, or could not have been reported without the system that is built in this paper. They have also provided a variable importance analysis of what kinds of behaviors were evident in the agents as they completed these benchmark tasks.

Reviewer AkiS
The authors provided a rationale for organizing the benchmark, and this seems sound. This is a long and exhaustive paper that is akin to collecting a dataset. Future work will ideally study specific questions using this data. The rebuttal also includes some new results on the rank correlation of model accuracy across different benchmarks.

Reviewer Uqtf
The authors calculated the proposed “distance to the Pareto frontier” on the data which leads to some interesting insights.

Reviewer YJJ9
The authors have reiterated that their insights in the paper are precise manifestation of folk wisdom and therefore valuable (even they might not be “surprising” to some readers). They updated the Appendix to provide more technical details.

**Reviewer Scores:**

Reviewer n4gN
NA

Reviewer nPVd
Said that they will maintain their current score, 8.

Reviewer AkiS
The reviewer said that they will maintain their score of 6 because they think it is more of an engineering system paper than a more standard ICLR paper that picks and solves a research problem.

Reviewer Uqtf
The response was adequate. I suspect that the score would have increased to a 6.

Reviewer YJJ9
The reviewer argued that if this is an engineering paper, then it needs to provide sufficient clarify in terms of user manuals, reproducible codebase etc. for users to adopt it. They maintain the original score. But after this comment, the authors provided the entire code repository to an anonymous link. So I believe that this comment is more or less addressed. I suspect the score might improve, at least slightly, to a 4.

---

### Decision · Program_Chairs · 2026-01-26

Accept (Poster)